# Tamsulosin use in benign prostatic hyperplasia and risks of Parkinson's disease, Alzheimer's disease and mortality: An observational cohort study of elderly Medicare enrollees

**Kin Wah Fung**[ID]*, **Fitsum Baye, Seo H. Baik**[ID]**, Clement J. McDonald**

Lister Hill National Center for Biomedical Communications, National Library of Medicine, National Institutes of Health, Bethesda, Maryland, United States of America

* kfung@mail.nih.gov

## Abstract

### Purpose

To study the effects of benign prostatic hyperplasia treatments, namely: alpha-adrenergic receptor blockers, 5-alpha-reductase inhibitors and phosphodiesterase-5 inhibitors on the risk of Parkinson's disease, Alzheimer's disease and mortality.

### Materials and methods

All male Medicare enrollees aged 65 or above who were diagnosed with benign prostatic hyperplasia and received one of the study drugs between 2007–2020 were followed-up for the three outcomes. We used Cox regression analysis to assess the relative risk of each of the outcomes for each study drug compared to the most prescribed drug, tamsulosin, while controlling for demographic, socioeconomic and comorbidity factors.

### Results and conclusions

The study analyzed 1.1 million patients for a mean follow-up period of 3.1 years from being prescribed one of the study drugs. For all outcomes, patients on tamsulosin were used as the reference for comparison. For mortality, alfuzosin was associated with 27% risk reduction (HR 0.73, 95%CI 0.68–0.78), and doxazosin with 6% risk reduction (HR 0.94, 95%CI 0.91–0.97). For Parkinson's disease, terazosin was associated with 26% risk reduction (HR 0.74, 95%CI 0.66–0.83), and doxazosin with 21% risk reduction (HR 0.79, 95%CI 0.72–0.88). For Alzheimer's disease, terazosin was associated with 27% risk reduction (HR 0.73, 95%CI 0.65–0.82), and doxazosin with 16% risk reduction (HR 0.84, 95%CI 0.76–0.92). Tadalafil was associated with risk reduction (27–40%) in all 3 outcomes. More research is needed to elucidate the underlying mechanisms of these observations. Given the availability of safer alternatives for treating benign prostatic hyperplasia, caution should be exercised

**Data Availability Statement:** Data availability statement The Medicare claims data belong to CMS, and we do not have permission to publish the raw data. CMS does not let us download (or distribute) any patient level data. The data stay on their machine, and we analyze it with software they provide on their machine. If researchers wish to access the raw data, they can contact the CMS Virtual Research Data Center https://resdac.org/cms-virtual-research-data-center-vrdc. However, data access requires the payment of a fee.

**Funding:** This research was supported by the Intramural Research Program of the NIH, National Library of Medicine. I confirm that the funders had no role in study design, data collection and analysis, decision to publish, or preparation of the manuscript. This research did not receive support from individual grants.

**Competing interests:** The authors have declared that no competing interests exist.

when using tamsulosin in elderly patients, especially those with an increased risk of developing neurodegenerative diseases.

## Introduction

Benign prostatic hyperplasia (BPH) is the most common benign neoplasm in aging men. The incidence starts to rise at age 40–45 years, reaching 60% by age 60 and up to 90% in the ninth decade [1–3]. Men with BPH can be asymptomatic. Management generally begins with behavioral modification, followed by medications, before considering more invasive options. According to the American Urological Association (AUA) guidelines, the mainstay of medical therapy for BPH includes alpha-adrenergic receptor blockers (AB), 5-alpha-reductase inhibitors (5-ARI) and phosphodiesterase-5 inhibitors (PDE5i) [4,5]. ABs are often used as the first-line treatment. Among them, tamsulosin is the most frequently prescribed, although most studies found no difference in efficacy or tolerability among ABs [4–7]. According to our analysis of Medicare data, over the past 15 years, tamsulosin use increased exponentially and has become the de facto medical treatment of choice (Fig 1).

However, recent studies have shown concerning results, linking tamsulosin with increased risk of Parkinson's disease (PD) and dementia [8–13]. One potential mechanism involves energy metabolism. Mitochondrial dysfunction has been hypothesized to be a cause of neurogenerative diseases, such as PD and Alzheimer's disease (AD) [14,15]. Among the elderly and patients with neurodegenerative diseases, there is impaired cerebral glucose metabolism, reduced mitochondrial biogenesis and lower adenosine triphosphate (ATP) levels [16–18]. ABs used in the treatment of BPH present a unique opportunity to test this hypothesis.

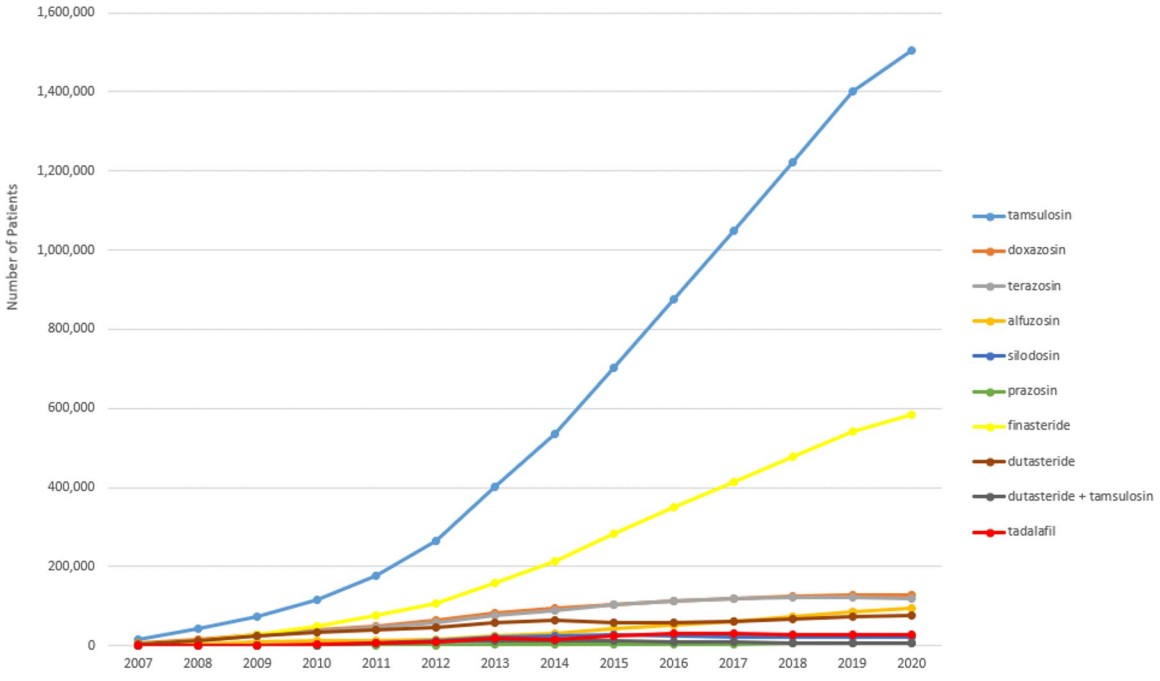

**Fig 1. Medical treatment for benign prostatic hyperplasia among Medicare enrollees ≥ 65 from 2007–2020.**

Terazosin, doxazosin and alfuzosin share a structural motif that binds and potentially increases the activity of phosphoglycerate kinase 1, the first ATP-generating enzyme in glycolysis. This will enhance glucose metabolism and may protect against neurodegenerative diseases [19]. Since tamsulosin lacks this structural relation, it has no effect on glucose metabolism. Several retrospective studies have found that, compared to terazosin/doxazosin/alfuzosin, tamsulosin is associated with increased risk of PD and dementia [9–13]. Another possible mechanism linking tamsulosin to dementia is tamsulosin's higher selectivity for the alpha-1A subtype of adrenoreceptors, which are more prevalent in the brain. In animal studies, suppressed expression of the alpha-1A adrenoceptors is linked to poor cognitive performance [20].

We report results from an observational cohort analysis among elderly male Medicare enrollees who were diagnosed with and received medical treatment for BPH, examining the association of drug use with the risks of PD, AD and mortality.

## Materials and methods

### 1. Study population and exclusion criteria

We included all male Medicare patients aged 65 or older who were diagnosed with and received medical therapy (see below for list of study drugs) for BPH between 2007 and 2020. Using the Virtual Research Data Center (VRDC) of the Centers for Medicare and Medicaid Services (CMS) [21], we accessed all available de-identified records of Medicare Part A (hospital insurance, 1999–2020), B (medical insurance, 1999–2020), C (Medicare Advantage, 2015–2020), and D (prescription drug insurance, 2007–2020) claims data for 100% Medicare beneficiaries. We focused on Medicare beneficiaries aged 65 or older, since younger Medicare beneficiaries typically need qualifying disability conditions to enroll, making them less representative of the general population. Patients with BPH were identified using the predefined chronic condition indicator in the CMS Chronic Conditions Data Warehouse [22]. To prevent prevalent user bias [23], we restricted our analysis to incident (new) users by excluding patients who had prescriptions for the study drugs within a washout period of 90 days (median length of supply of most study drugs) from their initial Medicare Part D enrollment. For the PD and AD analyses, we excluded patients diagnosed with these conditions before starting the study drugs. To ensure adequate follow-up and minimize incidental associations, we excluded patients with less than 1 year of follow-up after starting the drugs, similar to Simmering et al. [9] This study was declared not human subject research by the Office of Human Research Protection at the National Institutes of Health and by the CMS's Privacy Board. De-identified patient data was accessed via the CMS's VRDC enclave between 1 June 2023 and 15 October 2023. Researchers had no access to information that could identify individual participants during or after data collection.

### 2. Medical therapy

We identified the study drugs using the World Health Organization's Anatomical Therapeutic Chemical (ATC) Classification. We used generic names to find prescriptions for the study drugs in the Medicare outpatient drug (Part D) data. For AB, we included tamsulosin, alfuzosin, doxazosin, terazosin, silodosin and prazosin. Although prazosin is an older AB primarily indicated for hypertension, it can also be used to relieve BPH symptoms [24]. For 5-ARIs, we included dutasteride and finasteride. For PDE5i, we included tadalafil, the only drug in this class approved by the US Food and Drug Administration (FDA) for the treatment of BPH. We tracked the total days of drug exposure to study the cumulative effect (dose-response) and conducted a sensitivity analysis on patients with more than 1 year of cumulative exposure. To isolate the effect of individual drugs and ensure mutually exclusive treatment groups, we

censored patients who were on more than one drug (e.g., tamsulosin-dutasteride combination) and when they switched between study drugs.

## 3. Outcomes

We focused on 3 outcomes: death (all causes), PD and AD. We used International Classification of Diseases (ICD) codes to identify PD: ICD-9-CM code *332.0* before 1 October 2015 and ICD-10-CM code *G20-* thereafter. We used the CMS predefined chronic condition indicator to identify AD [22].

## 4. Comorbidities and socioeconomic factors

We identified 44 comorbidities (chronic conditions) with >1% prevalence in our study population, as tracked in the Chronic Conditions Data Warehouse. We used demographics and socioeconomic indicators (age, sex, race, rural residence, dual eligibility and low-income subsidy) as included in the Medicare data. For race, we used the race classification in the VRDC database: American Indian or Alaska Native, Asian, Black, Hispanic, Other, Unknown and White. We combined American Indian or Alaska Native, Other and Unknown into Other because of the small numbers. We included race as a covariate because race has been shown to affect the outcomes.

## 5. Statistical analysis

We followed all patients from the start of their study drugs until death, diagnosis of PD or AD, switching to a Medicare Advantage plan before 2015 (as data from these plans were only available since 2015), disenrollment from Medicare (Parts A, B or D), switching to another study drug, or reaching December 31, 2020, whichever came first. To mitigate selection bias among the treatment groups, we used a multiple logistic regression to calculate the propensity score (PS) for receiving any study drug other than tamsulosin, our reference drug. The PS represented the likelihood of receiving a drug other than tamsulosin, conditional on patient's age, race, rural residence, dual eligibility, low-income subsidy, and 44 chronic comorbidities at the start of follow-up (i.e., time-fixed). To ensure unbiased estimation of drug effects, we used the doubly robust method as in Funk et al. by running Cox regressions with PS as an additional adjustment, and including the same patient characteristics in PS calculation but treating them as time-varying covariates [25].

We used SAS Enterprise Guide Version 7.15 (SAS 9.4), SAS Institute Inc., Cary, NC, USA and conducted 2-sided test at a significance level of 0.05. This study is reported as per the Strengthening the Reporting of Observational Studies in Epidemiology (STROBE) guideline [26].

## Results

### The cohorts and baseline characteristics

Among 12,086,102 male Medicare enrollees aged 65 and above, 4,677,377 (38.7%) were ever diagnosed with BPH, of which 2,027,280 (16.8%) were started on one of the study drugs. After excluding patients diagnosed with BPH after the study, those with PD or AD before the study, and those with less than one year of follow-up, the study retained over 1.1 million patients in all three cohorts (Fig 2).

The characteristics of the mortality cohort are shown in Table 1. Among the study drugs, tamsulosin was the most frequently used (80%) followed by finasteride (6.4%), doxazosin (3.7%) and terazosin (3.7%). Overall, the median follow-up was 3.1 years, totaling over 4

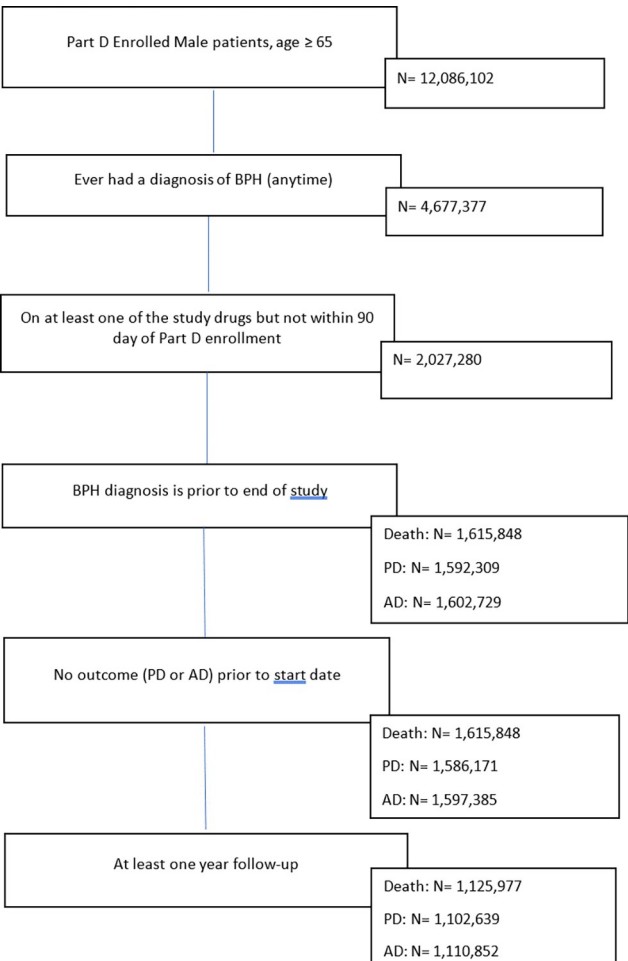

**Fig 2. Participants inclusion and exclusion flow chart for the 3 cohorts (BPH–benign prostatic hyperplasia; PD–Parkinson's disease; AD–Alzheimer's disease).**

million patient-years. The longest follow-up was observed in patients treated with tadalafil (median 3.8 years) and shortest with alfuzosin (median 2.7 years). Patients taking tamsulosin were the oldest (mean age 69.1) while those taking dutasteride were the youngest (mean age 67.8). The median treatment duration varied from 204 days (silodosin) to 616 days (doxazosin). The patient characteristics of the PD and AD cohorts were very similar to the mortality cohort.

## Effects of drug treatment on the outcomes

The effects of drug treatment are summarized in Table 2. Tamsulosin is used as reference to compare with all other treatments, since it is the most prescribed drug. For mortality risk, compared to tamsulosin, alfuzosin was associated with a significant 27% lower risk (hazard ratio (HR) 0.73, 95%CI 0.68–0.78), and doxazosin with a significant 6% lower risk (HR 0.94, 95%CI 0.91–0.97). No significant associations were observed for the other ABs. Tadalafil use was associated with the largest reduction in mortality risk, at 40% (HR = 0.60, 95%CI 0.55–0.66), while the 5-ARIs were associated with 7–24% reductions in risk.

For the risk of PD, compared to tamsulosin, terazosin was associated with a significant 26% risk reduction (HR 0.74, 95%CI 0.66–0.83), and doxazosin with a significant 21% reduced risk

**Table 1. Characteristics of patients in the mortality cohort.**

| | ALL | Tamsulosin | Alfuzosin | Doxazosin | Terazosin | Silodosin | Prazosin | Dutasteride | Finasteride | Tadalafil |
|---|---|---|---|---|---|---|---|---|---|---|
| N | 1,126,048 | 898,523 | 26,408 | 42,131 | 41,590 | 10,957 | 2,859 | 14,594 | 72,276 | 17,280 |
| % | 100.0% | 80% | 2.3% | 3.7% | 3.7% | 1.0% | 0.3% | 1.3% | 6.4% | 1.5% |
| Years of follow up: median(total) | 3.1 (4,008,070) | 3.1 (3,153,970) | 2.7(86,926) | 3.3(165,456) | 3.3(157,461) | 3.4(42,384) | 2.8(9,248) | 3.3(59,578) | 3.2(263,864) | 3.8(69,182) |
| Age at start of treatment: | | | | | | | | | | |
| Mean (SD) | 68.9(3.0) | 69.1(3.0) | 68.5(3.0) | 68.3(2.9) | 68.3(2.9) | 68.3(2.6) | 69.0(3.0) | 67.8(2.8) | 68.7(3.0) | 68.2(2.6) |
| 65–69 (%) | 679,201 (60.3) | 527,196(58.7) | 17,192 (65.3) | 28,814(68.5) | 28,241(68.0) | 7,521(69.1) | 1,645(59.0) | 11,023(75.9) | 45,484(63.0) | 12,085 (70.2) |
| 70–74 (%) | 390,849 (34.7) | 323,224(36.0) | 8,142(30.9) | 11,792(28.0) | 11,939(28.8) | 3,163(29.1) | 1,000(35.9) | 3,130(21.6) | 23,644(32.7) | 4,815(28.0) |
| 75–79 (%) | 55,927(5.0) | 48,031(5.3) | 1,003(3.8) | 1,454(3.5) | 1,339(3.2) | 202(1.9) | 142(5.1) | 371(2.6) | 3,077(4.3) | 308(1.8) |
| Days on treatment: | | | | | | | | | | |
| Median (IQR) | 385.0(90.0– 814.0) | 414.0(98.0– 883.0) | 385.0(90.0– 814.0) | 616.0(232.0– 1,224.0) | 598.0(190.0– 1,209.0) | 204.0(60.0– 614.0) | 210.0 (58.0– 617.0) | 443.5(150.0– 958.0) | 580.0(240.0– 1,127.0) | 264.0(87.0– 603.0) |
| < 1 year (%) | 12,694(1.1) | 414,061(46.1) | 12,694 (48.2) | 14,058(33.4) | 14,706(35.4) | 6,729(61.8) | 1,680(60.3) | 6,385(44.0) | 24,964(34.6) | 10,507 (61.1) |
| ≥ 1 year (%) | 13,643(1.2) | 484,390(53.9) | 13,643 (51.8) | 28,002(66.6) | 26,813(64.6) | 4,157(38.2) | 1,107(39.7) | 8,139(56.0) | 47,241(65.4) | 6,701(38.9) |
| Race/ethnicity | | | | | | | | | | |
| White (%) | 847,828 (75.3) | 678,760(75.5) | 21,141 (80.3) | 29,027(69.0) | 28,159(67.8) | 8,782(80.7) | 2,092(75.1) | 11,455(78.9) | 55,405(76.7) | 13,007 (75.6) |
| Black (%) | 84,535(7.5) | 67,360(7.5) | 1,374(5.2) | 4,370(10.4) | 3,574(8.6) | 705(6.5) | 276(9.9) | 884(6.1) | 4,662(6.5) | 1,330(7.7) |
| Hispanic (%) | 101,955(9.1) | 79,767(8.9) | 1,615(6.1) | 5,383(12.8) | 5,639(13.6) | 509(4.7) | 235(8.4) | 1,007(6.9) | 6,047(8.4) | 1,753(10.2) |
| Asian (%) | 40,375(3.6) | 32,227(3.6) | 669(2.5) | 1,534(3.6) | 2,348(5.7) | 324(3.0) | 76(2.7) | 516(3.6) | 2,475(3.4) | 206(1.2) |
| Other (%) | 51,284(4.6) | 40,337(4.5) | 1,538(5.8) | 1,746(4.2) | 1,799(4.3) | 566(5.2) | 108(3.9) | 662(4.6) | 3,616(5.0) | 912(5.3) |
| Ever Dual (%) | 159,139 (14.1) | 132,716(14.8) | 1,609(6.1) | 7,378(17.5) | 6,421(15.5) | 839(7.7) | 584(21.0) | 1,475(10.2) | 6,946(9.6) | 1,171(6.8) |
| Non-Dual LIS (%) | 24,294(2.2) | 19,939(2.2) | 351(1.3) | 1,070(2.5) | 1,002(2.4) | 180(1.7) | 75(2.7) | 212(1.5) | 1,283(1.8) | 182(1.1) |
| Non-Dual Non-LIS (%) | 942,544 (83.7) | 745,796(83.0) | 24,377 (92.6) | 33,612(79.9) | 34,096(82.1) | 9,867(90.6) | 2,128(76.4) | 12,837(88.4) | 63,976(88.6) | 15,855 (92.1) |
| Living in rural area (%) | 257,414 (22.9) | 210,151(23.4) | 4,111(15.6) | 9,834(23.4) | 8,552(20.6) | 1,695(15.6) | 690(24.8) | 3,181(21.9) | 16,401(22.7) | 2,799(16.3) |
| Days on treatment: median (IQR) | 385.0(90.0– 814.0) | 414.0(98.0– 883.0) | 385.0(90.0– 814.0) | 616.0(232.0– 1,224.0) | 598.0(190.0– 1,209.0) | 204.0(60.0– 614.0) | 210.0 (58.0– 617.0) | 443.5(150.0– 958.0) | 580.0(240.0– 1,127.0) | 264.0(87.0– 603.0) |
| < 1 year (%) | 12,694(1.1) | 414,061(46.1) | 12,694 (48.2) | 14,058(33.4) | 14,706(35.4) | 6,729(61.8) | 1,680(60.3) | 6,385(44.0) | 24,964(34.6) | 10,507 (61.1) |
| ≥ 1 year (%) | 13,643(1.2) | 484,390(53.9) | 13,643 (51.8) | 28,002(66.6) | 26,813(64.6) | 4,157(38.2) | 1,107(39.7) | 8,139(56.0) | 47,241(65.4) | 6,701(38.9) |

(HR 0.79, 95%CI 0.72–0.88). No significant associations were observed for the other ABs. Among the 5-ARIs, finasteride was associated with 10% risk reduction (HR 0.90, 95%CI 0.83–0.98), while dutasteride showed no significant effect. Tadalafil was associated with a significant 27% risk reduction (HR 0.73, 95%CI 0.61–0.87).

For the risk of AD, compared to tamsulosin, terazosin was associated with a significant 27% risk reduction (HR 0.73, 95%CI 0.65–0.82), and doxazosin with a significant 16% reduced risk (HR 0.84, 95%CI 0.76–0.92). No significant associations were observed for the other ABs and 5-ARIs. Tadalafil was associated with a significant 34% risk reduction (HR 0.66, 95%CI 0.54–0.80).

**Table 2. Cox regression results of drug treatment on the three outcomes (not shown here are the 44 chronic comorbidities that are included as covariates, N–total number of patients; E–patient with outcome event; HR–hazard ratio; CI–confidence interval; LIS–low-income subsidy).**

| | | Death | | Parkinson's disease | | Alzheimer's disease | |
|---|---|---|---|---|---|---|---|
| Patient count | | N = 1,125,977; E = 79,553 | | N = 1,102,639; E = 11,814 | | N = 1,110,852; E = 11,928 | |
| Index variable | Reference | N (%) | HR (95% CI) | N (%) | HR (95% CI) | N (%) | HR (95% CI) |
| *Alpha-adrenergic receptor blocker* | | | | | | | |
| Alfuzosin | Tamsulosin | | | | | | |
| -overall | | 26,337(2.3) | 0.73(0.68,0.78) | 25,880(2.3) | 0.90(0.79,1.03) | 26,087(2.3) | 0.88(0.76,1.01) |
| - >1 year usage | | 13,643(1.2) | 0.66(0.60,0.73) | 13,412(1.2) | 0.84(0.69,1.02) | 13,510(1.2) | 0.89(0.72,1.09) |
| Doxazosin | Tamsulosin | | | | | | |
| -overall | | 42,060(3.7) | 0.94(0.91,0.97) | 41,501(3.8) | 0.79(0.72,0.88) | 41,576(3.7) | 0.84(0.76,0.92) |
| - >1 year usage | | 28,002(2.5) | 0.90(0.86,0.94) | 27,658(2.5) | 0.75(0.66,0.85) | 27,710(2.5) | 0.73(0.64,0.83) |
| Terazosin | Tamsulosin | | | | | | |
| -overall | | 41,519(3.7) | 0.97(0.94,1.01) | 40,961(3.7) | 0.74(0.66,0.83) | 41,205(3.7) | 0.73(0.65,0.82) |
| - >1 year usage | | 26,813(2.4) | 0.93(0.88,0.97) | 26,487(2.4) | 0.68(0.59,0.78) | 26,587(2.4) | 0.69(0.60,0.80) |
| Silodosin | Tamsulosin | | | | | | |
| -overall | | 10,886(1.0) | 0.93(0.86,1.01) | 10,661(1.0) | 0.97(0.80,1.17) | 10,779(1.0) | 1.08(0.90,1.30) |
| - >1 year usage | | 4,157(0.4) | 0.94(0.82,1.08) | 4,071(0.4) | 0.87(0.64,1.19) | 4,107(0.4) | 1.16(0.87,1.55) |
| Prazosin | Tamsulosin | | | | | | |
| -overall | | 2,787(0.2) | 0.90(0.78,1.03) | 2,676(0.2) | 1.02(0.75,1.39) | 2,678(0.2) | 0.71(0.50,1.03) |
| - >1 year usage | | 1,107(0.1) | 1.12(0.92,1.37) | 1,059(0.1) | 1.23(0.77,1.95) | 1,056(0.1) | 0.80(0.45,1.41) |
| *5-Alpha-reductase inhibitor* | | | | | | | |
| Dutasteride | Tamsulosin | | | | | | |
| -overall | | 14,524(1.3) | 0.76(0.71,0.83) | 14,319(1.3) | 1.03(0.88,1.20) | 14,391(1.3) | 0.94(0.80,1.11) |
| - >1 year usage | | 8,139(0.7) | 0.74(0.66,0.83) | 8,029(0.7) | 0.95(0.76,1.18) | 8,056(0.7) | 0.97(0.77,1.22) |
| Finasteride | Tamsulosin | | | | | | |
| -overall | | 72,205(6.4) | 0.93(0.90,0.96) | 70,814(6.4) | 0.90(0.83,0.98) | 71,358(6.4) | 0.98(0.90,1.06) |
| - >1 year usage | | 47,241(4.2) | 0.91(0.87,0.95) | 46,302(4.2) | 0.91(0.82,1.00) | 46,662(4.2) | 0.90(0.81,0.99) |
| *Phosphodiesterase-5 inhibitor* | | | | | | | |
| Tadalafil | Tamsulosin | | | | | | |
| -overall | | 17,208(1.5) | 0.60(0.55,0.66) | 16,973(1.5) | 0.73(0.61,0.87) | 17,099(1.5) | 0.66(0.54,0.80) |
| - >1 year usage | | 6,701(0.6) | 0.60(0.52,0.70) | 6,606(0.6) | 0.76(0.57,1.01) | 6,676(0.6) | 0.59(0.42,0.83) |
| Age: 70–74 | 65–69 | 390,849(34.7) | 1.30(1.28,1.33) | 380,357(34.5) | 1.35(1.29,1.41) | 383,586(34.5) | 1.65(1.58,1.73) |
| 75–79 | 65–69 | 55,927(5.0) | 1.66(1.59,1.73) | 53,870(4.9) | 1.53(1.36,1.72) | 54,169(4.9) | 2.15(1.92,2.40) |
| Race*: Black | White | 84,535(7.5) | 0.95(0.93,0.98) | 83,336(7.6) | 0.76(0.70,0.82) | 82,837(7.5) | 1.29(1.21,1.38) |
| Hispanic | White | 101,955(9.1) | 0.79(0.77,0.82) | 100,186(9.1) | 0.83(0.77,0.89) | 100,031(9.0) | 1.18(1.10,1.25) |
| Asian | White | 40,375(3.6) | 0.71(0.68,0.75) | 39,698(3.6) | 0.86(0.77,0.97) | 39,981(3.6) | 0.83(0.74,0.94) |
| Other | White | 51,284(4.6) | 0.93(0.89,0.97) | 50,233(4.6) | 0.94(0.85,1.03) | 50,853(4.6) | 0.83(0.74,0.93) |
| Income: Dual eligibility | Non-dual, no LIS | 159,139(14.1) | 1.45(1.43,1.48) | 153,946(14.0) | 1.04(0.98,1.11) | 152,890(13.8) | 1.54(1.46,1.62) |
| Non-dual; on LIS | Non-dual, no LIS | 24,294(2.2) | 1.30(1.25,1.36) | 23,932(2.2) | 0.88(0.77,1.01) | 24,088(2.2) | 1.08(0.96,1.22) |
| Rural residence | Non rural | 257,414(22.9) | 1.19(1.17,1.21) | 252,004(22.9) | 0.98(0.94,1.03) | 253,855(22.9) | 1.04(0.99,1.09) |

* Race information was as recorded in the Medicare database. The original classification was American Indian or Alaska Native, Asian, Black, Hispanic, Other, Unknown and White. We combined American Indian or Alaska Native, Other and Unknown into Other because of the small numbers in these categories.

The significant risk reductions observed in all three outcomes were more pronounced with cumulative drug exposure of over 1 year.

Table 3 presents the incidence rates of the outcomes for assessment of absolute risk reduction. For mortality risk, the largest absolute reduction compared to tamsulosin was observed with alfuzosin (incidence 10.2 vs. 21.3, a 52% reduction) and tadalafil (7.8 vs. 21.3, a 63%

**Table 3. Number of events and incidence rate of the three outcomes.**

| Treatment group | Death | | Parkinson's disease | | Alzheimer's disease | |
|---|---|---|---|---|---|---|
| | Number of events | Event/1000 patient-year | Number of events | Event/1000 patient-year | Number of events | Event/1000 patient-year |
| All | 79,553 | 19.8 | 11,814 | 3.0 | 11,928 | 3.0 |
| Alpha-adrenergic receptor blocker | | | | | | |
| Tamsulosin | 67,239 | 21.3 | 9,744 | 3.2 | 9,934 | 3.2 |
| Alfuzosin | 889 | 10.2 | 220 | 2.6 | 186 | 2.2 |
| Doxazosin | 3,218 | 19.4 | 391 | 2.4 | 422 | 2.6 |
| Terazosin | 2,536 | 16.1 | 321 | 2.1 | 314 | 2.0 |
| Silodosin | 621 | 14.7 | 128 | 3.1 | 123 | 2.9 |
| Prazosin | 215 | 23.2 | 42 | 4.7 | 31 | 3.5 |
| 5-Alpha-reductase inhibitor | | | | | | |
| Dutasteride | 612 | 10.3 | 164 | 2.8 | 141 | 2.4 |
| Finasteride | 3,683 | 14.0 | 664 | 2.6 | 655 | 2.5 |
| Phosphodiesterase-5 inhibitor | | | | | | |
| Tadalafil | 540 | 7.8 | 140 | 2.1 | 122 | 1.8 |

reduction). For PD, the largest reduction was observed with terazosin and tadalafil (both 2.1 vs. 3.2, a 34% reduction). For AD, the largest reduction was observed with terazosin (2.0 vs. 3.2, a 38% reduction) and tadalafil (1.8 vs. 3.2, a 44% reduction).

## Discussion

The most significant finding of our study is that among the ABs used for the treatment of BPH, tamsulosin, the most commonly used drug, is associated with a higher risk of mortality (compared to alfuzosin and doxazosin), as well as higher risks of PD and AD (both compared to doxazosin and terazosin). Our results generally corroborate earlier studies, albeit our sample is much larger e.g., 4 times that of Simmering et al. and 18 times that of Duan et al. [9,13] We believe ours is the first to specifically examine the effect of BPH drug treatment on mortality. Compared to tamsulosin, alfuzosin is associated with a 27% reduction of mortality risk. More-over, effects on neurodegenerative diseases and mortality appear to be independent of each other. For example, terazosin has effects on neurodegenerative diseases but not mortality, while the reverse is observed for alfuzosin. More research is needed to elucidate the underlying mechanisms of these observed associations.

ABs are the mainstay of medical treatment for BPH. The smooth muscle in the prostate responds to alpha-adrenergic stimulation. Blocking the (predominantly alpha-1) adrenoceptors reduces bladder outlet resistance. Prazosin, the first selective alpha-1 blocker available, has demonstrated effectiveness in the relief of BPH symptoms but requires twice-daily dosing. Doxazosin and terazosin, also selective alpha-1 blockers, are administered once daily but require a titration period to a full therapeutic dose. Tamsulosin, administered once daily without the need for titration, has gained popularity due to this convenience, coupled with a lower incidence of side effects such as orthostatic hypotension. However, it is important to note that most studies indicate comparable efficacy and generally good tolerance among all ABs. In addition, alfuzosin, which our study associates with reduced mortality risk, is also dosed daily and does not require titration.

The incidence of neurodegenerative diseases has skyrocketed in recent decades [27,28]. Given that patients with BPH are typically elderly, they are at increased risk for

neurodegenerative diseases. In the light of our findings and other studies, it would be prudent to reconsider whether tamsulosin remains the optimal first-line treatment, given its association with increased risk of mortality and neurodegenerative diseases, despite its dosing convenience and fewer minor side effects. For patients with the appropriate indications (e.g., erectile dysfunction), 5-ARI or PDE5 should be considered as safer alternatives to tamsulosin.

We acknowledge the following limitations in our study. Like other observational studies, our study is subject to misclassification errors. Not all cases of BPH, PD and AD may be accurately documented in claims data. Drug exposure was inferred from prescription records without verification of actual dispensation or consumption. By design, to isolate the effect of individual drugs, we sanctioned patients who switched between drugs or were on combination drugs (e.g., dutasteride with tamsulosin). We did not include less frequently prescribed drug treatment options, such as anticholinergics and beta-3 adrenergic agonists. In our regression analysis, we controlled for demographic, socioeconomic and comorbidity factors, but there may be additional confounding variables not included in our regression models. Patients on tamsulosin in our study were generally older than those on other drugs, but this potential bias was mitigated by propensity score adjustment and including age as a covariate in our regression model.

## Conclusion

Based on our analysis of 1.1 million elderly male Medicare enrollees who were diagnosed with and received drug treatment for BPH, tamsulosin was associated with a higher risk of mortality compared to alfuzosin and doxazosin, and a higher risk of PD and AD compared to doxazosin and terazosin. Dutasteride and finasteride, compared with tamsulosin, were associated with a reduced risk of mortality, and tadalafil a reduced risk of mortality, PD and AD. Further research is needed to elucidate the underlying mechanisms of these observations. Given the availability of safer alternatives, caution should be exercised when using tamsulosin in elderly patients, particularly those at increased risk of developing neurodegenerative diseases.

## Author Contributions

**Conceptualization:** Kin Wah Fung, Fitsum Baye, Seo H. Baik, Clement J. McDonald.

**Data curation:** Fitsum Baye, Seo H. Baik.

**Formal analysis:** Kin Wah Fung, Fitsum Baye, Seo H. Baik.

**Funding acquisition:** Clement J. McDonald.

**Investigation:** Kin Wah Fung, Fitsum Baye, Seo H. Baik, Clement J. McDonald.

**Methodology:** Kin Wah Fung, Fitsum Baye, Seo H. Baik, Clement J. McDonald.

**Project administration:** Clement J. McDonald.

**Resources:** Clement J. McDonald.

**Supervision:** Clement J. McDonald.

**Writing – original draft:** Kin Wah Fung.

**Writing – review & editing:** Kin Wah Fung, Fitsum Baye, Seo H. Baik, Clement J. McDonald.

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
