## [Decision Letter · Decision Letter 0]

12 Jul 2024

PONE-D-24-16897Tamsulosin use in benign prostatic hyperplasia and risks of Parkinson disease, Alzheimer disease and mortality: An observational cohort study of elderly Medicare enrolleesPLOS ONE

Dear Dr. Fung,

Thank you for submitting your manuscript to PLOS ONE. After careful consideration, we feel that it has merit but does not fully meet PLOS ONE’s publication criteria as it currently stands. Therefore, we invite you to submit a revised version of the manuscript that addresses the points raised during the review process.

We look forward to receiving your revised manuscript.

Kind regards,

Stanisław Jacek Wroński, M.D., Ph.D, FEBU

Academic Editor

PLOS ONE

Journal Requirements:

3. Thank you for stating the following financial disclosure: ""This research was supported by the Intramural Research Program of the NIH, National Library of Medicine.""

Reviewers' comments:

Reviewer's Responses to Questions

**Comments to the Author**

1. Is the manuscript technically sound, and do the data support the conclusions?

Reviewer #1: Yes

2. Has the statistical analysis been performed appropriately and rigorously? 

Reviewer #1: Yes

3. Have the authors made all data underlying the findings in their manuscript fully available?

Reviewer #1: Yes

4. Is the manuscript presented in an intelligible fashion and written in standard English?

Reviewer #1: No

5. Review Comments to the Author

Reviewer #1: I would like to congratulate the authors for the novel study that explores the risk of PD among AB drug users. However, I 've noticed a few issues within the manuscript which are listed as follows:

1. The flow and language of the manuscript is fair. But I believe the language and grammar can be brushed up further to improve the quality.

2. This manuscript misses to discuss a recently published meta-analysis on the same question: Lamichhane P, Tariq A, Akhtar AN, Raza M, Lamsal AB, Agrawal A. Risk of Parkinson's disease among users of alpha-adrenergic receptor antagonists: a systematic review and meta-analysis. Ann Med Surg (Lond). 2024 May 6;86(6):3409-3415. doi: 10.1097/MS9.0000000000002117. PMID: 38846867; PMCID: PMC11152853.

3. Please add statement regarding ethical approval from ethical review committee and need of consent from the study subjects.

6. PLOS authors have the option to publish the peer review history of their article (what does this mean?). If published, this will include your full peer review and any attached files.

Reviewer #1: No

---

## [Author Response · Author response to Decision Letter 0]

25 Jul 2024

Dear PLOS ONE Editors,

As authors of the manuscript “Tamsulosin use for benign prostatic hyperplasia associated with increased risk of Parkinson Disease, Alzheimer Disease and mortality” we would like to thank the editors and reviewers for their thoughtful comments. We believe we have addressed all the concerns as detailed below.

Comments to the Author

1. Is the manuscript technically sound, and do the data support the conclusions?

Reviewer #1: Yes

2. Has the statistical analysis been performed appropriately and rigorously? 

Reviewer #1: Yes

3. Have the authors made all data underlying the findings in their manuscript fully available?

Reviewer #1: Yes

4. Is the manuscript presented in an intelligible fashion and written in standard English?

Reviewer #1: No

>> the manuscript has been revised by professional copyeditors to improve the readability and quality of writing

5. Review Comments to the Author

Reviewer #1: I would like to congratulate the authors for the novel study that explores the risk of PD among AB drug users. However, I 've noticed a few issues within the manuscript which are listed as follows:

1. The flow and language of the manuscript is fair. But I believe the language and grammar can be brushed up further to improve the quality.

>> the manuscript has been revised by professional copyeditors to improve the readability and quality of writing

2. This manuscript misses to discuss a recently published meta-analysis on the same question: Lamichhane P, Tariq A, Akhtar AN, Raza M, Lamsal AB, Agrawal A. Risk of Parkinson's disease among users of alpha-adrenergic receptor antagonists: a systematic review and meta-analysis. Ann Med Surg (Lond). 2024 May 6;86(6):3409-3415. doi: 10.1097/MS9.0000000000002117. PMID: 38846867; PMCID: PMC11152853.

>> Thanks for the suggestion. The said reference, together with 2 other relevant references have been added and discussed. 

3. Please add statement regarding ethical approval from ethical review committee and need of consent from the study subjects.

>> This has been included in the manuscript: “This study was declared not human subject research by the Office of Human Research Protection at the National Institutes of Health and by the CMS’s Privacy Board.” The documentation of the determination has been updated separately through the submission website.

---

## [Decision Letter · Decision Letter 1]

8 Aug 2024

Tamsulosin use in benign prostatic hyperplasia and risks of Parkinson’s disease, Alzheimer’s disease and mortality: An observational cohort study of elderly Medicare enrollees

PONE-D-24-16897R1

Dear Dr Kin Wah Fung

We’re pleased to inform you that your manuscript has been judged scientifically suitable for publication and will be formally accepted for publication once it meets all outstanding technical requirements.

Kind regards,

Stanisław Jacek Wroński, M.D., Ph.D, FEBU

Academic Editor

PLOS ONE

---

## [Editor Report · Acceptance letter]

13 Aug 2024

PONE-D-24-16897R1 

PLOS ONE

Dear Dr. Fung, 

I'm pleased to inform you that your manuscript has been deemed suitable for publication in PLOS ONE. Congratulations! Your manuscript is now being handed over to our production team.

Kind regards, 

on behalf of

Dr. Stanisław Jacek Wroński 

Academic Editor

PLOS ONE